# Retinal Vasculature in Schizophrenia Spectrum Disorder

**DOI:** 10.3390/bioengineering13010035

**Published:** 2025-12-28

**Authors:** Caroline Simon Sherman, Erik Gunnarsson, Nycole Hidalgo, Victoria Chen, Kevin Zhang, Shuo Chen, Hwiyoung Lee, Hugh O’Neill, L. Elliot Hong, Osamah Saeedi

**Affiliations:** 1University of Maryland School of Medicine, Baltimore, MD 21201, USAfhidalgo@som.umaryland.edu (N.H.); 2Department of Ophthalmology and Visual Sciences, University of Maryland School of Medicine, Baltimore, MD 21201, USA; 3Department of Pathology, Johns Hopkins University, Baltimore, MD 21218, USA; 4Division of Biostatistics and Bioinformatics, Department of Epidemiology and Public Health, University of Maryland School of Medicine, Baltimore, MD 21201, USAhwiyoung.lee@som.umaryland.edu (H.L.); 5Maryland Psychiatric Research Center, Department of Psychiatry, University of Maryland School of Medicine, Baltimore, MD 21201, USA; honeill@som.umaryland.edu; 6Department of Psychiatry & Behavioral Sciences, University of Texas Health Science Center at Houston, Houston, TX 77054, USA

**Keywords:** optical coherence tomography angiography, retinal vasculature, schizophrenia

## Abstract

The purpose of this research is to determine whether retinal vasculatures differ between participants with schizophrenia spectrum disorder (SSD) and controls. Ninety participants (51 SSD, mean age 35.8 ± 13.5, and 39 controls, mean age 35.5 ± 11.4) underwent 3 × 3 mm^2^ macular and 6 × 6 mm^2^ optic nerve head (ONH) optical coherence tomography angiography (OCTA) scans. En face macula and ONH region images were divided into quadrants, binarized, and then skeletonized. Skeletonized vessel densities were compared between our two groups. Additionally, the foveal avascular zone (FAZ) size and acircularity index were compared between the two groups. There was significantly decreased vessel density in the temporal region of the ONH in the SSD group compared to controls (*p* = 0.033). Interestingly, the decreased vessel density was already present in patients with SSD in younger adulthood as compared to the controls (*p* = 0.006). There were no significant group differences in vessel density in any other region of the ONH, the ONH overall, any region of the macula, or the macula overall. There were also no significant group differences in the FAZ size or acircularity index. These data suggest there may be abnormal peripapillary retinal vasculature in patients with SSD. Whether this is a specific ocular vascular deficit or related to more systemic vascular abnormalities in SSD remains to be determined.

## 1. Introduction

Schizophrenia spectrum disorders (SSDs) are a group of debilitating psychiatric disorders that affect approximately 24 million people worldwide [1]. Individuals with SSD have an increased risk of premature mortality with a much shorter life expectancy compared to the U.S. general population [2]. While schizophrenia may be broadly characterized by positive symptoms (e.g., delusions, hallucinations), negative symptoms (e.g., apathy, social withdrawal), and cognitive deficits, the etiopathophysiology remains obscure and is still under intense investigation.

Prior research has shown that the disease is associated with alterations in cerebral microvasculature [3]. Further, schizophrenia is associated with higher vascular co-morbidities and adverse vascular events [4,5]. Given this association, it is postulated that there is a significant vascular component to the pathophysiology of schizophrenia and some researchers even postulate that brain microvascular dysfunction is the core pathophysiology in SSD [6,7,8,9].

The retina, which can be thought of as “the window to the brain,” is a highly metabolically active and vascularized neural tissue that is the only component of the central nervous system that can be easily imaged directly in vivo, providing a unique opportunity to study the neural microvasculature. Various tests have been employed to analyze structural and functional retinal changes, including an electroretinogram (ERG) and optical coherence tomography (OCT), both of which can be easily achieved during an outpatient visit. It has been shown in our group and others that patients with schizophrenia have a significantly reduced retinal nerve fiber layer (RNFL) and macular thickness [10,11]. Furthermore, a longer duration of illness is associated with thinner macular and peripapillary RNFL thicknesses [10]. The severity of positive symptoms has also been associated with a smaller macular volume [12]. Some retinal structural abnormalities extend to healthy first-degree family members of people with schizophrenia [13]. Beyond retinal structural differences, we found ten reports that have studied the retinal vasculature in patients with schizophrenia, showing overall that patients with schizophrenia show alterations in the retinal vasculature as compared to controls [14,15,16,17,18,19,20,21,22,23]. Patients with schizophrenia have wider venules, narrower arterioles, greater vascular tortuosity, and more complex vascular branching [14]. Altogether, this suggests that the retinal microvasculature may be a valuable biomarker for schizophrenia. However, retinal microvascular density declines with normal aging [24,25]. Hence, it remains unclear as to whether the retinal microvascular anomalies identified in patients with SSD is part of their abnormal aging process, or if they already emerge in younger patients.

OCT angiography (OCTA) is a relatively new office-based modality that allows for the visualization and quantification of the retinal microvasculature. Prior research has shown a decreased retinal vascular density in neurodegenerative diseases such as Alzheimer’s and Parkinson’s disease [26,27,28]. This study aims to comprehensively investigate the relationship between OCTA-based vascular biomarkers and SSD. We also aim to investigate the role of age in the effects of schizophrenia on the retinal vasculature. The ultimate aim of this study is to assess the potential use of OCTA-based vascular biomarkers in SSD using currently available technology. To that end, we compared multiple metrics using different image analysis techniques in both the peripapillary and macular regions in patients with SSD and control subjects. Metrics assessed included vessel density, as well as the size and acircularity index of the foveal avascular zone (FAZ).

## 2. Materials and Methods

### 2.1. Subjects

This study was conducted between 2017 and 2020. The study was approved by the Institutional Review Board at the University of Maryland, and subjects provided written informed consent after receiving an explanation of the nature and possible consequences of the study. All methods were carried out in accordance with relevant guidelines and regulations at the University of Maryland. Participants with SSD were recruited from the Maryland Psychiatric Research Center outpatient clinics and neighboring mental health clinics. SSD included a diagnosis of either schizophrenia or schizoaffective disorder, which was confirmed based on the Diagnostic and Statistical Manual of Mental Disorders, fifth edition (DSM-5). Controls were recruited by media advertisement. Controls were excluded if they had any current DSM-5 Axis I diagnoses or family history of psychotic illness in first-degree relatives. Participants were excluded from analysis if they had pre-existing ocular pathologies, such as macular pathologies (age-related macular degeneration, epiretinal membrane, macular hole), retinopathies (retinal vascular occlusion, retinal dystrophy), glaucoma, optic neuropathy, uveitis, previous intraocular surgery (except uncomplicated cataract surgery), and penetrating ocular trauma. Participants with diabetic macular edema on OCT or with self-reported diabetic eye disease were excluded. Participants were also excluded if there was a history of major neurological or unstable medical illness, substance abuse within the last month, or substance dependence within the last three months (except smoking and marijuana use). Figure 1 outlines images that were excluded in this study. Analysis of various demographic factors of the included and excluded participants was conducted to investigate any significant differences in the excluded cohort. The ages of the included and excluded participants were compared with a *t*-test. Sex, race, ethnicity, and diagnosis of the included and excluded participants were compared with chi-square tests.

### 2.2. OCTA Measurements

Patients underwent OCTA with a Cirrus 5000 AngioPlex scanner from Carl Zeiss Meditec (Jena, Germany). The scanner acquires data at a scan depth of 2 mm, with an axial resolution of 5 μm, at a rate of approximately 27,000 axial scans (A-scans) per second for structural OCT and 68,000 for OCTA. Both 3 × 3 mm^2^ and 6 × 6 mm^2^ images centered on the macula and optic nerve head (ONH) were taken. Images that were decentered or noted to have movement artifact were excluded after review by one trained grader (EG). This led to the exclusion of 26 eyes in the ONH set, 83 eyes in the macula set, and 45 eyes in the FAZ set (Figure 1).

### 2.3. Image Analysis

The Fiji application of ImageJ (NIH, Bethesda, MD, USA) was used to process the OCTA images. Based on the literature support and extensive subjective quality assessments of the Phansalkar, Niblack, and Otsu local binarization methods, we selected the Phansalkar method for macula images and Niblack for the ONH images [29,30,31]. Next, binarized images were skeletonized, making each vessel 1 pixel wide to assess capillary density and to minimize the effect of larger vessels on vessel density.

ONH: Vessel density. We assessed the full retinal thickness and superficial vascular plexus captured in the 6 × 6 mm^2^ ONH angiograms. We did not assess the deep vascular plexus due to poor image quality. The Early Treatment of Diabetic Retinopathy Study (ETDRS) grid was overlaid to allow regional analysis in 4 quadrants divided into inner and outer parts, for a total of 8 regions: inner superior, outer superior, inner nasal, outer nasal, inner inferior, outer inferior, inner temporal, and outer temporal (Figure 2) [32]. The ETDRS grid was manually centered over the ONH using the translate function within ImageJ to ensure alignment of the grid and the image. Two individuals analyzed the ONH data. They were trained on a dataset of 5 images prior to analyzing all of the images. The coefficient of variation between the two graders’ manual centering of images was 0.0057 (ICC(2, 1) = 0.9996)). Data included vessel density values for all 8 regions, as well as the average vessel density across the 8 regions.

Macula: Vessel density. We assessed the whole retina slabs and superficial vascular plexus captured in the 3 × 3 mm^2^ angiograms, in line with previous OCTA studies analyzing macular images [31,33]. The ETDRS grid was overlaid to allow regional analysis in the 4 inner regions: superior, nasal, inferior, and temporal (Figure 3) [32,34,35,36]. The ETDRS grid was manually centered over the FAZ using the translate function within ImageJ to ensure alignment. One individual graded the macula data and each image was reviewed by the principal investigator (OS). Data included vessel density values for all 4 regions, as well as the average vessel density across the 4 quadrants.

FAZ size and acircularity index. We measured the FAZ size and acircularity index using the 3 × 3 mm^2^ whole retina images. The perimeter of the FAZ was manually outlined with the freehand selection tool within ImageJ (Figure 4) [37,38,39]. For any image where there was a questionable segment of the FAZ, a second experienced grader (OJS) was consulted and the 6 × 6 mm^2^ whole retina image from the same eye was referenced to guide the FAZ tracing. Any image for which no FAZ could be determined due to image quality or artifact was excluded. Once the FAZ was outlined, the FAZ area and perimeter were obtained from ImageJ. Using these two values, we were able to calculate an acircularity index as defined by Tam et al.—the ratio of the perimeter of the FAZ to the perimeter of the circle with the same area as the FAZ [40].

### 2.4. Statistical Analysis

We first compared the demographic variables between the control and SSD groups using the two-sample *t*-test for continuous variables and chi-square test for categorical variables. Because of their cited effect on vessel density, we also compared the signal strength intensity (SSI) and axial length between the two groups using a *t*-test [41,42,43,44].

For the ONH data, linear mixed effects models were used to examine the main effect of diagnosis on vessel density. Specifically, laterality (left and right eye), quadrant (4 per eye), and ring (inner and outer) were used as the repeated measurements. The inner and outer rings were collapsed if no significant ring-based effect was found. The linear mixed-effects model was adjusted for age and sex. We also investigated the role of age by splitting our sample into older (age > 30) and younger (age ≤ 30) groups and analyzing the effect of diagnosis on vessel density using the same model previously discussed. An age of 30 was chosen as a cutoff in concordance with past retinal vasculature research dividing participants by age [23]. Finally, we used a statistical mixed model to analyze the correlation between the ONH vessel density and RNFL thickness in each quadrant between participants with SSD and controls, and a separate sub-analysis stratified by age, separated into younger (<30) and older (≥30) age groups.

Macula vessel density data were similarly analyzed with mixed effects models except where laterality (left and right eye) and quadrant (4 per eye) were repeated measures. For both ONH and macula data, a contrast test was used for post hoc testing, which tests the difference in vessel density between controls and participants with SSD at each quadrant. For adjustment of *p*-values for multiple comparisons, the Benjamini–Hochberg Procedure was used to control the false discovery rate.

The effect of diagnosis on the FAZ size and acircularity index was analyzed with a linear regression model. All analyses were performed using R v4.1.1 software (R Foundation for Statistical Computing). Specifically, we used the ‘lmer’ function in the ‘lme4’ library to implement linear mixed-effects models.

## 3. Results

An analysis of differences in demographic factors of included and excluded participants was conducted. For the ONH images, the average age of the included participants was significantly higher than that of the excluded participants (t = −2.10, *p*-value = 0.041). There were significantly more females in the excluded group compared to the included group (chi-square = 7.33, *p*-value = 0.007). There was no significant difference in race (chi-square = 5.27, *p*-value = 0.261), ethnicity (chi-square = 1.12, *p*-value = 0.290), or diagnosis (chi-square = 2.23, *p*-value = 0.135) between the included and excluded participants in terms of the ONH images. For the macula images, there was no significant difference in age (t = −1.04, *p*-value = 0.302), sex (chi-square = 0.032, *p*-value = 0.858), race (chi-square = 3.68, *p*-value = 0.451) or ethnicity (chi-square = 0.423, *p*-value = 0.516), or diagnosis (chi-square = 0.084, *p*-value = 0.772) between the included and excluded participants. For the FAZ images, there was no significant difference in age (t = −1.49, *p*-value = 0.139), sex (chi-square = 0.845, *p*-value = 0.358), race (chi-square = 0.987, *p*-value = 0.912) or ethnicity (chi-square = 0.178, *p*-value = 0.674), or diagnosis (chi-square = 0.491, *p*-value = 0.484) between the included and excluded participants.

Table 1 outlines the demographics for the included participants. There were more males in the SSD group than the control group (*p*-value = 0.001). Ages and race demographics were not significantly different between the two groups. There was no significant difference in the axial length of both the ONH and macula images or the SSI of the ONH images between the control and SSD groups. There was a difference in the SSI of the macula images between the control and SSD groups. This is because the SSI values in the control group were all 10, resulting in no variation in the data. As a result, the test was highly sensitive, potentially leading to the rejection of the null hypothesis even when the difference is very small.

### 3.1. ONH Vessel Density by Eight ETDRS Regions

The full mixed model showed no significant main effect on diagnosis but showed a significant effect of diagnosis by temporal region interaction (*p* = 0.033) (Table 2 and Table 3, Figure 5). Therefore, the ring data were collapsed, and we performed post hoc tests on the significant interaction: there was a significantly decreased vessel density in the temporal region of the ONH of participants with SSD compared to controls. There were no significant differences found between any of the other ETDRS regions or in the ONH overall (Figure 6 and Figure 7). The results of the superficial plexus-specific statistical analysis are available in the Appendix A.

When data were split by age (older group age >30, younger group ≤ 30), there was a significant decrease in vessel density in the temporal quadrant of participants with SSD compared to controls in the younger group (*p* = 0.0066). There was no significant difference in the vessel density in the temporal quadrant between the controls and participants with SSD in the group of participants over 30 (*p* = 0.6537) (Figure 8). When age was modeled as a continuous variable, a diagnosis-by-age interaction was observed for temporal ONH vessel density, indicating that the diagnosis effect varied as a function of age and was not dependent on a single age cutoff.

Given that previous studies have found an association between SSD diagnosis and RNFL thickness, we compared vessel density and RNFL thickness across the younger and older SSD participants and healthy controls. We found that the ONH vessel density and RNFL thickness have a significant negative correlation in SSD participants aged < 30 years old, specifically in the nasal (R= −0.46, *p* = 0.01) and inferior (R= −0.40, *p* = 0.03) quadrants. Alternatively, in SSD participants aged ≥ 30, there was a significant positive correlation between vessel density and the RNFL thickness in the superior quadrant (R = 0.32, *p* = 0.047). No significant relationships were demonstrated in the temporal quadrant in either age group in participants with SSD (*p* > 0.05). Across all quadrants and ages, the control group did not show a significant correlation between vessel density and RNFL thickness (*p* > 0.05). Appendix A details these results.

### 3.2. Macula Vessel Density

There were no significant differences in macula vessel density between the SSD and control groups in any quadrant or overall (Table 4 and Table 5).

Using the Niblack binarization technique for ONH images and the Phansalkar binarization technique for macula images, vessel densities in the binarized, non-skeletonized images were similarly analyzed. Using the same statistical analysis, there were no significant findings in either the ONH or macula images using these non-skeletonized images.

### 3.3. FAZ Size and Acircularity Index

The FAZ size (in mm^2^) and acircularity index were compared between the SSD and control groups. As is shown in Table 6, the average FAZ size for the schizophrenia group was 0.26 ± 0.14 and for the control group was 0.32 ± 0.11 (*p*-value = 0.38). The acircularity index for the SSD group was 1.35 ± 0.15 and for the control group was 1.32 ± 0.14 (*p*-value = 0.27). There was no significant difference in the FAZ size or acircularity index between the SSD and control groups.

## 4. Discussion

We present one of the largest studies to date investigating differences in retinal blood vessel density between subjects with SSD and control subjects. We found a significantly decreased vessel density in the temporal region of the ONH of patients with SSD compared to controls. Interestingly, after splitting our sample by age, this significant decrease in vessel density in the temporal ONH region was only seen in the younger SSD patients. Further, we found no significant differences between the FAZ size/acircularity index or macula vessel density in patients with SSD compared to the controls.

Axial length has been shown to significantly affect the vessel density and FAZ size and acircularity in the retina [42,43]. Because of this, we compared axial length in our control and SSD groups, finding no significant difference. The SSI of the images is another very important variable that can affect vessel density [41]. Studies have shown that the relationship between parafoveal capillary density and age is confounded by the SSI [44]. To mitigate this factor, not only did we use a poor SSI as an exclusion criteria, but we also ensured that the two groups did not have significantly different SSIs.

In general, this is consistent with the results of prior studies showing retinal vascular associations with SSD, with some notable differences. Silverstein et al. noted decreased vessel density in the left eyes of subjects with SSD and not the right [22]. With a larger sample, we found no significant difference in the eye (left vs. right) main effect or eye-by-group interactions in any of the vessel density measures.

In a study that included 12 subjects with SSD and 8 subjects with bipolar disorder, Koman-Wierdak et al. found significantly decreased vessel densities in the macular deep vascular complex in patients with schizophrenia compared to controls, but they did not find any differences in the peripapillary vessel density [19]. Notably, that study used a different device for the OCTA examination and the scanning areas of the macular and peripapillary scans differed from those seen in our investigation.

In a study of 30 subjects with SSD, Bannai et al. also examined the macular retinal vasculature using a swept source OCT. Contrary to the results of other studies cited here, they reported a higher overall skeletonized vessel density in the superficial slab of the maculae in the right eyes of patients with SSD [17].

In a study of 22 subjects with SSD, Budakoglu et al. found a significantly lower peripapillary vessel density specifically in the temporal quadrant of the OCTA images of participants with schizophrenia, but not overall [18]. We similarly found a significantly lower peripapillary vessel density specifically in the temporal region of participants with SSD compared to controls. It is unclear why only a reduced vessel density was seen in the temporal region; this is perhaps related to thinning of the temporal retinal nerve fiber layer that is found in subjects with SSD [11].

Finally, Silverstein et al. found the FAZ size to be significantly enlarged and retinal vascular density to be reduced in a study of 28 participants with schizophrenia [22]. We did not find a significant difference in the FAZ size of patients with SSD. This may be due to the difference in demographics as that population was majority Caucasian, whereas our study population included more ethnic-minority participants. There are known differences in the FAZ size between Caucasian and African Americans that could contribute to this difference [45]. Furthermore, that investigation reported data from only the superficial retinal layer, whereas our study’s analysis used entire retinal thickness. We also studied the FAZ acircularity index, another commonly used metric of vascular health, which was not associated with SSD diagnosis [43].

Unlike in these prior studies, we took advantage of the larger sample sizes and examined the age effect, and we found that the significantly lower vessel density in the temporal ONH region was already present in the younger SSD patients and, in fact, the group differences become insignificant in the older cohorts. It has been shown that retinal vessel density decreases with age in healthy populations [46]. We observed similar trends of a reduced vessel density with age in the healthy control group. However, in patients with SSD, this trend was less apparent. We hypothesize that SSD impacts the microvasculature at an early age, but it would paradoxically also render further age-related retinal-vessel-density reduction less likely given that a certain level of blood supply is mandatory to maintain the highly active visual functions supported by the retina in patients and controls alike. Longitudinal studies are needed to support or refute this ‘floor’ effect hypothesis. This may also explain some of the inconsistent findings in previously reported retina vasculature studies, where age ranges were not consistent across studies.

While this study benefits from being the largest known study of its kind, there were some noteworthy limitations. While we used Spectral Domain OCT, the current clinical standard for OCTA in the United States, there are faster devices that use Swept Source OCTA that may provide greater detail, although it is not clear whether this would improve the diagnostic ability of this modality. Furthermore, OCTA currently allows for the high structural resolution of capillaries, but does not give dynamic information about capillary velocities or flow rates. In addition, vascular density measures only one aspect of the microcirculatory function, and may miss other important blood-flow information such as vessel patency and permeability integrity. We found significant and promising associations in the whole retina but not in the specific plexuses, but note that the image quality in the superficial and deep plexuses was suboptimal. Although false-discovery-rate correction was applied, these exploratory, region-specific associations should still be interpreted with caution given the large number of correlated tests and the known sensitivity of OCTA metrics to image quality and segmentation effects. As expected, there were more males in the SSD cohort than in the control group, but this was accounted for in our analysis. Our methods utilized a semi-automated technique with some manual outlining and centering as well as automated processing on ImageJ to obtain the skeletonized vessel density data.

In conclusion, this study provides a more comprehensive understanding of the effects of schizophrenia on the retinal vasculature by assessing the macular and peripapillary retinal vasculature. Further research is needed to understand the mechanisms underlying the findings.

## Figures and Tables

**Figure 1 bioengineering-13-00035-f001:**
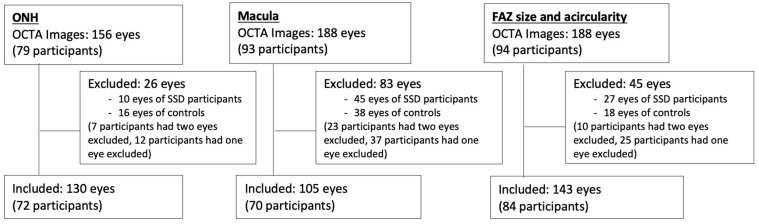
Image exclusion. Numbers of excluded images for each of the three image analysis groups: optic nerve head (ONH) vessel density, macula vessel density, and foveal avascular zone (FAZ) size and acircularity index.

**Figure 2 bioengineering-13-00035-f002:**
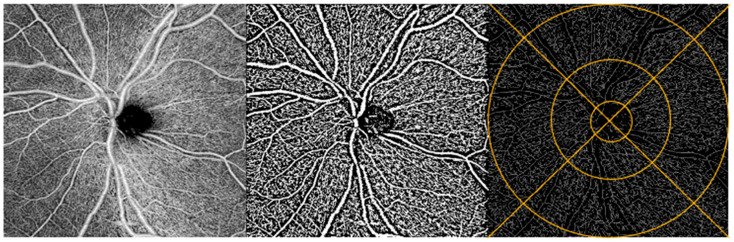
ONH image analysis. OCTA image of the ONH region of a participant’s retina (**left**). Same OCTA image with Niblack binarization applied (**middle**) and with binarization + skeletonization applied (**right**). Example of ETDRS grid overlay on skeletonized image (**right**).

**Figure 3 bioengineering-13-00035-f003:**
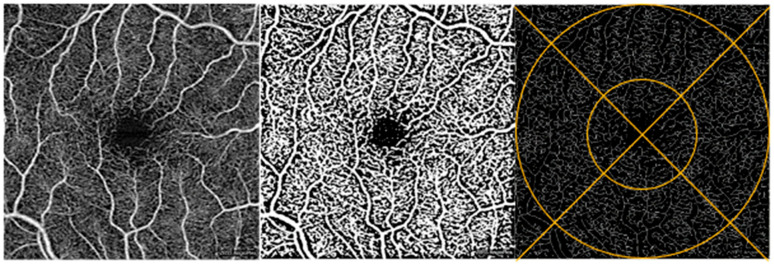
Macula image analysis. OCTA image of the macula of a participant’s retina (**left**). Same OCTA image with Phansalkar binarization applied (**middle**) and with binarization + skeletonization applied (**right**). Example of ETDRS grid overlay on skeletonized image (**right**).

**Figure 4 bioengineering-13-00035-f004:**
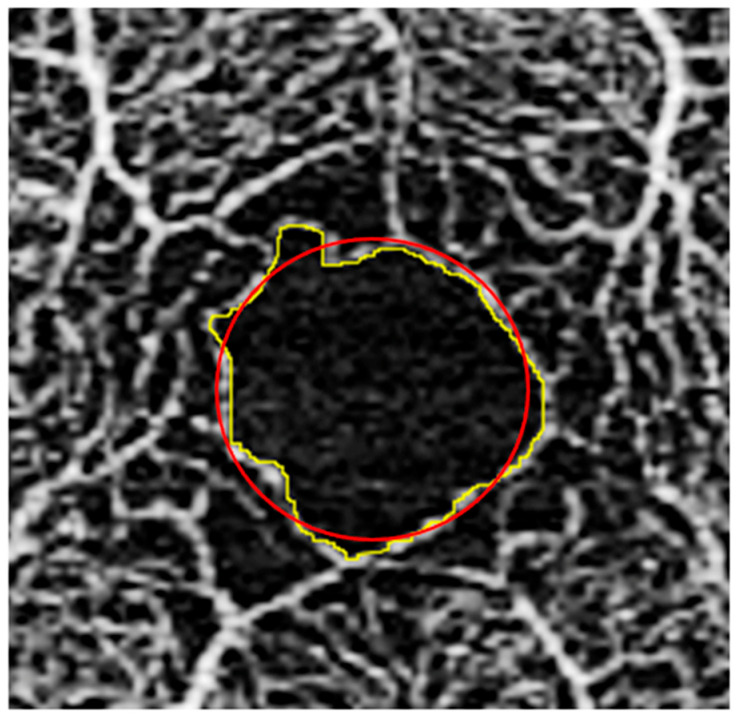
FAZ image analysis. Fovea with outline traced along border of FAZ (yellow); superimposed circle (red) shown, which was used for acircularity index calculation as defined by Tam et al. [40].

**Figure 5 bioengineering-13-00035-f005:**
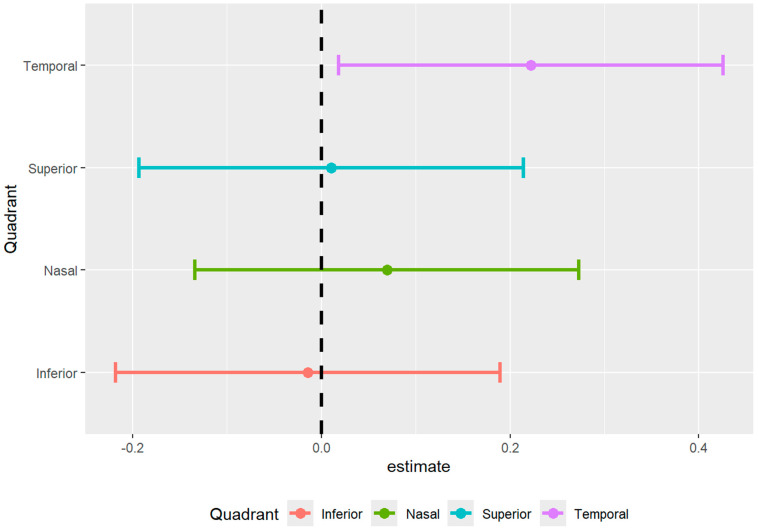
Forest plot of difference in density between SSD cohort and healthy controls.

**Figure 6 bioengineering-13-00035-f006:**
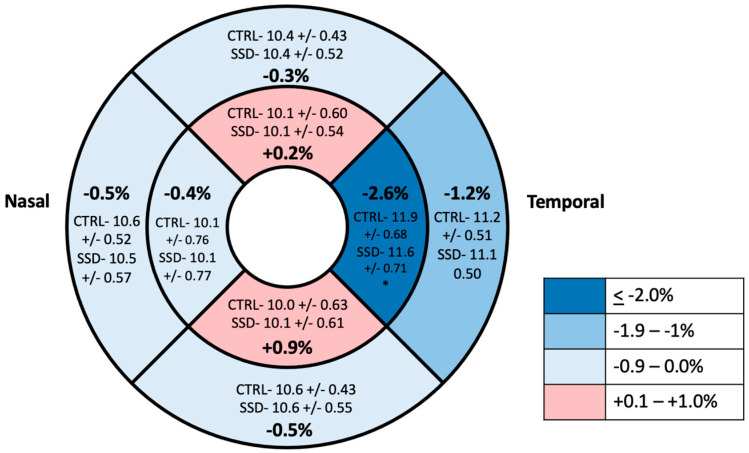
ETDRS grid with average vessel density in the 8 grid regions in the ONH region. Values are reported as means with percentage difference between mean in control and SSD groups. * Statistically significant difference.

**Figure 7 bioengineering-13-00035-f007:**
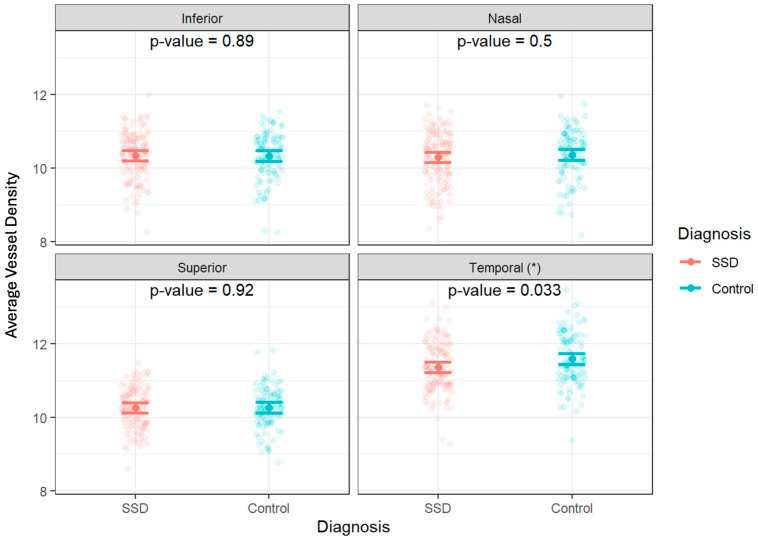
Individual optic-nerve-head vessel density measurements by diagnosis and quadrant. Data are shown as jittered dot plots with group means and 95% confidence intervals overlaid for control and schizophrenia spectrum disorder (SSD) groups. * A significant difference was observed in the temporal quadrant (*p* = 0.033).

**Figure 8 bioengineering-13-00035-f008:**
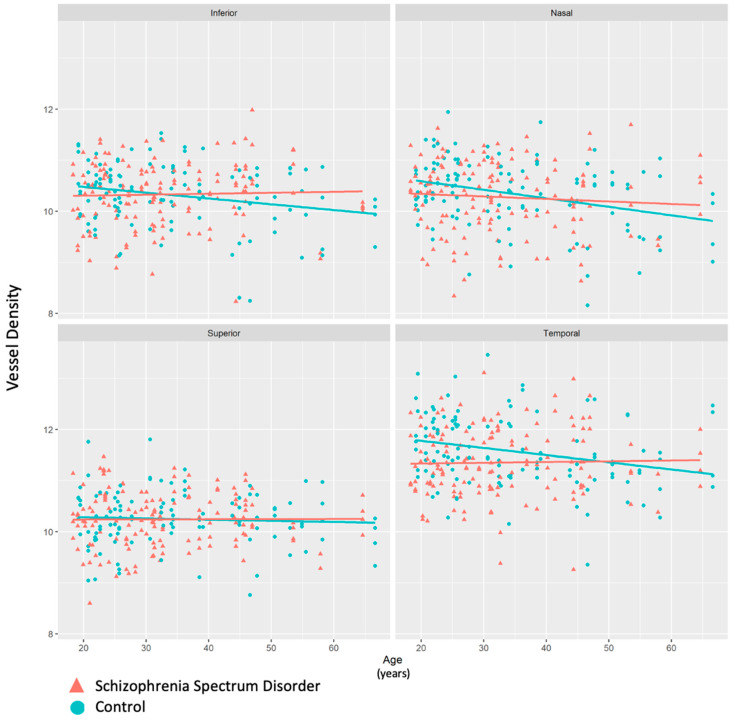
Scatterplot depicting vessel density values by age in each of the four quadrants in the ONH region. Participants with schizophrenia spectrum disorder (SSD) had significantly lower vessel density than controls, particularly in the younger age group.

**Table 1 bioengineering-13-00035-t001:** Cohort demographics.

Participants	Controls, N = 39	SSD, N = 51	*p*-Value
**Age (mean ± SD)**	35.83 +/− 13.52	35.55 +/− 11.39	0.9171
**Sex, n (%)**			0.0014
**Male**	17 (43.59%)	39 (76.47%)	
**Female**	22 (56.41%)	12 (23.53%)	
**Self-Identified Race, n (%)**			0.1333
**White**	18 (46.15%)	24 (47.06%)	
**Black**	12 (30.77%)	24 (47.06%)	
**Asian**	3 (7.69%)	1 (1.96%)	
**Native American**	2 (5.13%)	0 (0.00%)	
**Other**	4 (10.26%)	2 (3.92%)	
**Ethnicity, n (%)**			0.8127
**Non-Hispanic**	37 (94.87%)	50 (98.04%)	
**Hispanic**	2 (5.13%)	1 (1.96%)	
**Eyes**	**Controls, N = 72 eyes**	**SSD, N = 100 eyes**	
**Right**	35 (48.61%)	50 (50.00%)	
**Left**	37 (51.39%)	50 (50.00%)	

**Table 2 bioengineering-13-00035-t002:** Results of mixed linear model for ONH region by quadrant with false-discovery-rate-adjusted *p*-values. Note: Inferior quadrant was used as a reference.

	Regression Coefficient	*p*-Value
**Diagnosis × Nasal**	−0.084	0.395
**Diagnosis × Superior**	−0.025	0.799
**Diagnosis × Temporal**	−0.236	0.017

**Table 3 bioengineering-13-00035-t003:** Results of post hoc testing contrast (control–SSD).

Quadrant	Estimate	*p*-Value
Inferior	−0.014	0.889
Nasal	0.069	0.502
Superior	0.011	0.918
Temporal	0.222	0.033

**Table 4 bioengineering-13-00035-t004:** Results of mixed linear model for macula region by quadrant with false-discovery-rate-adjusted *p*-values. Note: Inferior quadrant was used as a reference.

	Regression Coefficient	*p*-Value
**Diagnosis × Nasal**	0.002	0.985
**Diagnosis × Superior**	0.109	0.260
**Diagnosis × Temporal**	0.127	0.189

**Table 5 bioengineering-13-00035-t005:** Results of post hoc testing contrast (control–SSD).

Quadrant	Estimate	*p*-Value
Inferior	0.043	0.777
Nasal	0.041	0.786
Superior	−0.065	0.668
Temporal	−0.083	0.585

**Table 6 bioengineering-13-00035-t006:** Average FAZ size (mm^2^) and acircularity index. Values are reported as mean ± standard deviation.

	Control, N = 39 (64 Eyes)	SSD, N = 45 (79 Eyes)	Regression Coefficient	*p*-Value
**FAZ Size (mm^2^)**	0.32 ± 0.11	0.26 ± 0.14	−0.019	*p*-value > 0.05 (0.350)
**Acircularity Index**	1.32 ± 0.14	1.35 ± 0.15	0.044	*p*-value > 0.05 (0.093)

## Data Availability

The datasets used and/or analyzed during the current study are available from the corresponding author on reasonable request.

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
