# Peer review of "Bioengineering2026, 13(1), 35;https://doi.org/10.3390/bioengineering13010035"

_bioengineering, 2025, doi:10.3390/bioengineering13010035_

Round 1

Reviewer 1 Report

Comments and Suggestions for Authors

This manuscript answers a question—whether retinal microvasculature differs between individuals with schizophrenia spectrum disorders (SSD) and healthy controls—using OCTA with careful image processing and mixed-effects modeling. The study’s main takeaways are clearly stated and easy to follow: peripapillary vessel density is lower in the temporal quadrant among SSD participants, a difference that is already apparent in younger adults, while macular density, FAZ size, and FAZ acircularity show no group differences. The age-stratified analyses add useful suggestions.

I encourage several clarifications and modest analytical refinements that will improve robustness and readability without changing the study’s scope. First, please report the final number of eyes per analysis set (ONH, macula, FAZ) after quality exclusions directly in the Results and tables; Figure 1 is helpful, but exact denominators alongside each model outcome will help readers judge precision and selection effects. Given that a substantial number of images were excluded for macula and FAZ, a brief comparison of included vs. excluded eyes/participants (now said to be in the supplement) should be summarized in the main text with any meaningful differences highlighted. Second, the sex imbalance between groups is expected in SSD but still material; you adjust for sex, which is appropriate, yet a sensitivity analysis stratified by sex or adding a sex-by-diagnosis term for the temporal quadrant would show whether the key finding is uniform. Similarly, although axial length and SSI are compared between groups, consider including SSI and axial length as covariates in the mixed models (or provide a matched-SSI sensitivity analysis), especially because macular SSI in controls shows ceiling values that remove variance and could confound negative findings there. Third, the age threshold of 30 is motivated by prior retinal work; nonetheless, readers would benefit from a complementary analysis modeling age continuously (diagnosis-by-age interaction) to confirm that the younger-onset effect is not an artifact of a single cutoff. Presenting partial effects or simple slopes would make this especially digestible.

On measurement and reproducibility, the manual centering of ETDRS grids and freehand FAZ tracing are reasonable but introduce subjectivity. The reported “coefficient of variation” for centering is encouraging, yet it is not an intuitive reliability metric for readers; please also report inter-grader agreement as an ICC or provide an example panel demonstrating the range of centering differences. Because macula grading relied on a single grader, a small duplicate-grading subset with agreement metrics would strengthen confidence. Your choice of Niblack vs. Phansalkar binarization is justified, but given that results from non-skeletonized images were null, a short sensitivity table showing whether the temporal-quadrant finding persists across alternative binarization parameters (or at least with/without skeletonization) would bolster claims of robustness.

The correlation analyses between ONH vessel density and RNFL thickness are intriguing, particularly the quadrant-specific sign reversals across age strata. Because a negative correlation between density and thickness is counterintuitive, a few sentences of mechanistic context or a cautionary note about multiple testing and image-quality influences would prevent over-interpretation. Wherever possible, accompany p-values with effect sizes and confidence intervals; the temporal-quadrant group difference that survives FDR (p≈0.03) is statistically modest, and communicating its magnitude in intuitive units (e.g., absolute and percent differences with 95% CIs) will help readers judge clinical meaning.

A few presentation and style points will further smooth the paper. Tighten the Abstract and Discussion to clearly separate what was hypothesized, what was found (with one sentence on effect size), and what remains speculative. The “floor-effect” idea is interesting—frame it explicitly as a hypothesis for longitudinal testing rather than a post-hoc explanation. Ensure consistent terminology (“systemic,” not “systematic,” when referring to vascular abnormalities), and run a careful language pass to fix minor typos and formatting artifacts typical of pre-production layouts. Finally, because the main positive finding is anatomically specific (temporal peripapillary), a schematic figure that overlays the significant region on a representative ONH skeleton map, plus a forest-style panel with model estimates and CIs for all quadrants, would make the result more visually persuasive at a glance.

To sum up, this is a well-designed and clinically relevant study with a clear, anatomically coherent signal in the temporal peripapillary retina among SSD—especially in younger adults. With the modest clarifications and sensitivity checks above, the paper will make a solid contribution to the literature and will be accessible to non-specialists in both psychiatry and ophthalmology.

Reviewer 2 Report

Comments and Suggestions for Authors

The authors studied the difference of retinal vasculature between schizophrenia patients and healthy controls. The manuscript shows a systematic analysis and solid statistical comparisons. The conclusion is solid and inspiring, indicating microvascular dysfunction could potentially be related to the pathology of schizophrenia. The paper can be published if these following issues can be revised:

Major issues:

  1. Figure 6: It is better to draw every single data point out to avoid losing the information of data distribution. These data points could be in a light color, whereas mean and SD could be in a dark color to distinguish them. It is better to report the exact p values of every panel, no matter significant or not.

Minor issues:

  1. Line 15, line 113, line 261: “2” in “3x3mm2” and “6x6mm2” should be superscript.
  2. Line 62: The brackets “[ ]” are missing in Ref 14 .
  3. Line 167: “Tam et al” should be labeled with the number of the reference so that readers can find it more easily.

Round 2

Reviewer 1 Report

Comments and Suggestions for Authors

good revisions